# Estimating Genetic Variability and Heritability of Morpho-Agronomic Traits of M5 Cowpea (*Vigna unguiculata* (L.) Walp) Mutant Lines

**DOI:** 10.3390/ijms26157543

**Published:** 2025-08-05

**Authors:** Kelebonye Ramolekwa, Motlalepula Pholo-Tait, Travis Parker, Goitseone Malambane, Samodimo Ngwako, Lekgari Lekgari

**Affiliations:** 1Department of Crop and Soil Sciences, Botswana University of Agriculture and Natural Resources, Private 0027, Gaborone 00267, Botswana; 201000256@buan.ac.bw (K.R.); gmalambane@buan.ac.bw (G.M.); sngwako@buan.ac.bw (S.N.); 2Faculty of Research and Graduate Studies, Research Centre for Bioeconomy, Botswana University of Agriculture and Natural Resources, Private Bag 0027, Gaborone 00267, Botswana; 3Department of Plant Sciences, University of California, Davis, CA 95616-8780, USA; trparker@ucdavies.edu; 4Department of Field Crops and Horticulture, National Agricultural Research Development Institute, Private Bag 00177, Gaborone 00267, Botswana; lekgari@nardi.org.bw

**Keywords:** cowpea, mutation breeding, mutant lines, heritability, genetic advance, genetic variability, phenotypic variation, correlation

## Abstract

Induced mutation plays an integral part in plant breeding as it introduces new variability among the population. A study was conducted in cowpeas [*Vigna unguiculata* (L.) Walp] to assess the yield divergence, heritability, genetic advance, and correlation among the M5 Tswana cowpea mutants. The experiment utilized seven genotypes under rainfed and supplementary irrigation during the 2022/23 and 2023/24 cropping seasons. The mutant lines demonstrated significant variations in days to 50% emergence (DE) and days to 50% flowering (DF). Tswana emerged earlier (5–7 days) and flowered in 21–54 days across the two seasons, compared to some of the mutant lines. The yield and yield components varied among some mutant lines and the control. Most importantly, mutants outperformed the Tswana control for some of these traits, indicating potential for genetic improvement. An analysis of genetic parameters revealed minimal environmental influences on some of the observed traits (GH, PN, GY), while others showed little environmental impact. Variation in heritability (H^2^) and genetic advance (GA%) between the two seasons limited the contribution of genotypic effects in the expression of the studied traits. Correlation analysis revealed strong and significant positive associations between DE and GH, as well as between DF and PW. Most traits, except DF and PW, were positively correlated with grain yield (GY), although the correlations were not significantly different. Cluster analysis grouped the genotypes into four distinct clusters. Principal component analysis (PCA) revealed the superiority of mutant lines (Tswana-300Gy-214, Tswana-400Gy mutant lines, and Tswana-500Gy-31) in their association with improved GY, pod weight (PW), 100-seed weight (100-SW), and seed number per pod (SN/P). Interestingly, the Tswana control formed a separate cluster and diverged from the mutants in PCA, suggesting that induced mutagenesis effectively targeted genes controlling the traits considered in this study.

## 1. Introduction

Genetic diversity refers to the extent of genetic variability within and among populations of a species [1]. It is a primary force of selection and evolution, as well as the genetic basis for breeders [2]. Given the pressing need to ensure food production in the face of an expanding human population and a changing environment, it is crucial to develop superior crop mutant lines [3]. This also necessitates proactive conservation and maintenance of genetic resources to ensure future breeding success [4]. In this context, improving cowpea’s [*Vigna unguiculata* (L.) Walp] genetic diversity is a relevant strategy, particularly in less-resourced communities where the crop is the most important food legume. It significantly contributes to the livelihoods of millions of smallholder farmers who depend on it for economic sustenance and nutritional health [4,5].

Cowpea is a predominant food legume of the *Vigna* species, cultivated over approximately 12 million hectares. It ranked as the third most significant legume crop [6]. The species is a globally recognized important diploid (2n = 22) leguminous crop [7] with narrow genetic diversity among domesticates. The narrow genetic diversity has been attributed to its self-pollinating nature, restricted evolution from wild germplasm, and limited gene transfer between wild and cultivated lines [5,8]. Therefore, utmost attention is required to adopt efficient, innovative plant breeding technologies to expand the genetic base of cowpea [9]. A key aspect of these breeding attempts focuses significantly on exploiting genetic variability in agro-morphological characteristics to expand the cowpea gene pool, as well as that of other crops [10]. This focus has led to the adoption of numerous strategies to broaden the genetic base of cowpea [11]. Among these strategies, induced mutation using gamma radiation has been successfully used to generate substantial genetic variability, allowing the generation of mutants with desired agronomic characteristics [12]. This is a technique mostly used compared to others due to its rapid genetic variation, cost-effectiveness, high energy level, and high penetration rate into target tissues, and is ubiquitously applicable, with broad applications [13].

Induced variability is a valuable approach for rapidly increasing overall genetic diversity, which is important for crop output [14]. Crop yield is a complex phenotypic trait influenced by multiple genes and environmental factors [12,15], which may be aided by an evaluation of its attributed traits for efficient breeding [12,16]. The interaction of different genes with additive effects on phenotypic variability often governs the inheritance of these traits. Hence, to estimate the success of selection in mutant populations, genetic parameters such as the genotypic coefficient of variation (GCV), phenotypic coefficient of variation (PCV), heritability (H^2^), and genetic advance (GA) are crucial [17,18]. These parameters quantify genetic and phenotypic variation, assisting breeders in establishing selection criteria [18]. The genotypic coefficient of variation and phenotypic coefficient of variation represent the magnitude of genetic and phenotypic variation, respectively, while heritability reveals the proportion of phenotypic variation due to genetics [19]. High heritability allows effective selection, whereas low heritability may result in challenging selection due to environmental influences, and it is more useful when considered in conjunction with genetic advance as a percentage [20]. PCV and GCV values greater than 20% are considered high, values between 10% and 20% are considered medium, and values below 10% are regarded as low [18]. According to Yadesa et al. [21], the heritability of a specific trait is considered very high when it is 80% or more, moderately high when it is 60–79%, medium when it is 40–59% and low when it is 40% or less. Keeping the aforementioned gaps and scopes in view, the present study was conducted to analyze the extent of genetic variability and heritability estimates of cowpea grain yield and its associated traits in a population derived from mutagenesis over two successive growing seasons.

## 2. Results

### 2.1. Combined Analysis of Variance Indicating Performance of the Mutant Lines Across the Two Seasons

Season and genotype (G) effects were highly significant (*p* ≤ 0.05) for all traits except for GH and PL. The genotype-by-season interaction effect was highly significant at *p* ≤ 0.05 for DE, DF, PW, SNP, and 100 SW. This indicates that genotypes differed significantly in their response to the season effects for these measured traits. The overall combined ANOVA results are presented in Table 1.

### 2.2. Performance of M5 Cowpea Mutants in Agronomic Traits Across the Two Seasons

The number of days to 50% emergence (Figure 1A) and the number of days to 50% flowering (Figure 1B) were significantly fewer (*p* < 0.05) in the 2022/23 season compared to the 2023/24 season for mutant lines. Among the mutant lines, Tswana-300Gy-202 and Tswana-300Gy-214 reached 50% emergence earlier within 5 days compared to the control (7 days) and other mutant lines in the 2022/23 season, while Tswana-500Gy-53 took longer (9 days) to emerge. In the 2023/24 season, the control showed early germination (9 days), but it was not significantly different from Tswana-500Gy-53 (10 days). Tswana-400Gy-85 had significantly delayed germination, reaching 50% germination in 13 days. However, it was not significantly different from Tswana-400Gy-49 (11 days) and Tswana-500Gy-31 (12 days).

Regarding the days to 50% flowering during the 2022/23 season, Tswana, along with Tswana-400Gy-49, Tswana-500Gy-31, and Tswana-500Gy-53, showed significantly shorter flowering times (*p* < 0.05), reaching 50% flowering within 21–22 days. The Tswana-300Gy mutant lines and Tswana-400Gy-85 had delayed flowering, reaching 50% flowering in 25 days. However, in the 2023/24 season, Tswana reached 50% flowering in 54 days, which was earlier than all mutant lines except for Tswana-300Gy-202, which reached 50% flowering in 55 days. Notably, Tswana consistently reached 50% flowering earlier across both seasons, whereas Tswana-400Gy-85 exhibited delayed 50% flowering in both seasons (Figure 1B). However, the growth habit (GH) showed no significant variation between the two seasons, indicating stability across the environment (Figure 1C). 

### 2.3. Yield and Yield Trait Performance of M5 Cowpea Mutants Across the Two Cropping Seasons 

The performance of the mutant lines showed significant improvement in the 2023/24 season compared to the 202223 season (Figure 2). During the 2023/24 season, Tswana-500Gy-53 had higher PN (31 pods), followed by the control variety, which also had much greater PN (28 pods) than most of the mutant lines. However, no significant variations were observed for PN among the cowpea lines in the 2022/23 season (Figure 2A). PW differed significantly (*p* < 0.05) among cowpea lines across the two seasons (Figure 2B). In the 2022/23 season, Tswana-300Gy-202 showed significantly higher PW (32.655 g) compared to the control (28.629 g), Tswana-300 Gy-214 and Tswana-500Gy-31. In the 2023/24 season, Tswana exhibited a significantly lower *(p<0.05)* PW (65.860 g) compared to all mutant lines. The highest significant PW was observed for Tswana-300Gy-214 (101.078 g), and this was significantly higher compared to Tswana-300Gy-202 (74.96 g), Tswana-400Gy-49 (86.518 g), Tswana-500Gy-31 (84.650 g), and Tswana-500Gy-53 (89.803 g). Seed number per pod (SN/P) significantly differed among the mutant lines in the 2022/23 season, with Tswana-400Gy-49 having significantly higher *(p* < 0.05) SN/P (12 seeds) than the control (9 seeds) and others. However, no significant variation in SN/P was observed in the 2023/24 season (Figure 2C). 

In the 2022/23 season, Tswana control showed a significantly (*p* < 0.05) lower 100-SW (10.116 g) compared to Tswana-300Gy-202 (11.719 g), Tswana-400Gy-49 (11.667 g), and Tswana-500Gy-31 (11.834 g) and a lower 100-SW (10.116 g) compared to Tswana-300Gy-202 (11.719 g), Tswana-400Gy-49 (11.667 g), and Tswana-500Gy-31 (11.834 g). A significantly lower 100-SW (13.270 g) was observed for the Tswana control, and this was significantly lower than that of mutant lines except Tswana-500Gy-53 (13.663 g) during the 2023/24 season (Figure 2D).

Grain yield (GY) showed insignificant variation between the mutant lines in the 2022/23 season. However, Tswana-500Gy-31 showed the highest GY (49.940 g), while Tswana-400Gy-49 recorded the lowest GY (34.563 g). In contrast, significant differences were observed for GY among the mutant lines in the 2023/24 season. Tswana showed significantly lower (*p* < 0.05) GY (94.335 kg/ha^−1^) compared to Tswana-400Gy-49 (111.479 kg/ha^−1^) and Tswana-500Gy-31 (122.039 kg/ha^−1^), whereas others did not differ significantly from the control (Figure 2E). Overall, Tswana showed consistently poor performance across several measured traits, while Tswana-500 Gy-31 performed reasonably well in terms of yield (Figure 2)

### 2.4. Genetic Parameters for the M5 Cowpea Population During the 2022/23 and 2023/24 Cropping Seasons

Variations in genetic parameters between the two seasons are presented in Table 2 and Table 3. Minimal environmental influence was observed across the two seasons, with most traits recording low GV and PV. However, a notable difference was observed for PW, which recorded a higher GV (151.038) and GY (60.178) in the 2023/4 season. Moreover, PV for GY was significantly high across both seasons (61.888 and 141.986), indicating significant genetic variation within the population. On the other hand, the GCV and PCV were generally low for most of the measured traits, suggesting narrow genetic variation for these traits. In the 2022/23 season, DE (17.452%) and GH (18.122%) recorded a moderate GCV. Similarly, in the 2023/24 season, GH and PW also recorded a moderate GCV. Regarding the PCV in the 2022/23 season, high values were observed for DE (20.296%), GH (38.553%), and PN (28.277%). A moderate PCV was observed for GY (19.580%), while other traits recorded low PCV values. The 2023/24 season showed a high PCV for GH (47.001%), while a moderate PCV was observed for DE (11.624%), PN (12.454%), PW (14.878%), and GY (11.347%).

H^2^ estimates revealed high-to-moderate values for most traits in both seasons, except for PN and GY, which had low H^2^ values of 5.6% and 8.4%, respectively, in the 2022/23 season, and SN/P (6.2%) in the 2023/24 season. Regarding GA, most traits recorded low values ranging from 0.376 for GH to 3.056 for DF in the 2022/23 season. In the 2023/24 season, low values were also recorded, ranging from 0.061 for SN/P to 4.696 for PN, except for Gy and PW, with moderate-to-high values of 10.404 and 24.337, respectively. In the 2022/23 season, GA% was high for DE (30.912%) and moderate for DF (13.467%) and GH (17.547%), while other traits recorded low GA%. In the 2023/24 season, GA% was high for GH (28.237%) and PW (28.322%), and moderate for DE (12.437%), PN (18.172%), and 100-SW (11.483%). Other traits had low GA%. Traits with low GA% indicate limited potential for genetic improvement through selection, while those with moderate-to-high GA% suggest a greater scope for developing improved cultivars.

### 2.5. Correlation Coefficient Analysis

Pearson correlation analysis revealed significant relationships among some measured traits. A high, positive and significant correlation was observed between DF and PW (r = 0.93), and DE and PN (r = 0.90). Additionally, a significantly high but negative correlation was observed between DF and PN (r = −0.86). The weakest correlation was observed between DE and GY (0.03), and was not statistically significant (*p* ≤ 0.05). Positive correlations, although not statistically significant, were observed between GY and traits such as DE, GH, PN, PL, SN/P, and 100-SW, while DF and PW showed a negative correlation with GY. The results are presented in Figure 3 below.

### 2.6. Agro-Morphological Cluster Analysis

Cluster analysis based on agro-morphological traits categorized the seven genotypes into four distinct clusters (Figure 4) through complete linkage analysis. The Tswana control did not cluster with any other mutant lines, while Tswana-300Gy-202 and Tswana-300Gy-214 were grouped together. Tswana-400Gy-49, Tswana-500Gy-31, Tswana-400Gy-85, and Tswana-500Gy-53 created two additional groups, respectively.

### 2.7. Principal Component Analysis

The principal component results are summarized in Table 4. The results showed that the first three principal components (PCs) with eigenvalues greater than one accounted for 89.46% of the overall variation. PC1, PC2, and PC3 accounted for 41.26%, 26.59%, and 21.61% of the overall variance, respectively. Only PC1 and PC2 were interpreted because they accounted for the majority of the variance in the analyzed data. The main contributors to PC1 were PN and PL, with an explained variation of 41.26%, while other traits revealed substantial negative involvement. DE, PN, and GH were the main contributors to PC2 with an explained variance of 26.59%.

The PCA biplot was used to visualize the performance of genotypes and their trait associations (Figure 5). PC1 and PC2 explained 58.54% of the total variation. The Tswana control was associated with a high pod number. Among the mutant lines, Tswana-300Gy-202 was associated with an increase in PL, while Tswana-300Gy-214 was related to high DF, PL, and 100 SW. Tswana-400Gy-85 was associated with high DF, PW, and 100SW, while Tswana-400Gy-49 was related to high SN/P, GH, and 100 SW. Moreover, Tswana-500 Gy-31 was associated with improved Gy and GH, while Tswana-500 Gy-53 was associated with improved PN, GH, and DE.

## 3. Discussion

### 3.1. Developmental Stages and Growth Habits

Induced mutation plays a significant role in the genetic diversity of crop species. Gamma irradiation has been shown to enhance genetic variability by yielding novel mutant lines with promising traits that contribute toward advancing agronomic and agricultural improvement [22]. In the current study, observations on the evaluation of developmental stages and growth habits revealed the effect of seasons on the performance of the genotypes. The mutant lines showed different performances across the two seasons. The Tswana-300Gy-202 and Tswana-300Gy-214 mutant lines demonstrated early emergence, likely due to their high seed quality. This quality encompasses factors such as vigor, viability, and moisture content. These characteristics directly influence cellular metabolic activity, which in turn affects seed germination and emergence [23,24]. Additionally, changes in the physiological and biochemical processes involved in seed germination and emergence may have contributed to the observed differences in days to 50% emergence among the mutant lines. The Tswana control consistently flowered earlier in both seasons, possibly due to specific inherent genetic alleles influencing its internal clock and hormonal pathways [25]. Environmental factors and potential stress tolerance may result in the observed differences between the mutant lines. Delayed flowering may be attributed to genetic differences created by the mutation that affect normal flowering mechanisms [26]. This appeared to be more prevalent in random mutagenesis, which was used in this study. This has been supported by various studies, which reported differences in days to 50% emergence due to random mutagenesis. Early emergence in cowpea mutants was observed by Kang et al. [27] and Victor et al. [28]. On the other hand, Rukesh et al. [29] and Thounaojam et al. [30] reported an increased number of days to 50% emergence in green gram. Other researchers also reported delayed flowering in cowpea lines [28,31]. On the contrary, an early flowering time has been observed in some cowpea mutants compared to their parent [32,33]. These conflicting results are likely due to the influence of several factors, including the specific cowpea mutant lines used, variations in environmental conditions across study locations, and differences in experimental design, such as the number of replications.

### 3.2. Yield and Yield Components

Optimizing cowpea production entails understanding the factors that influence both the crop’s developmental stage and yield. In grain legumes, grain yield is a complex trait, governed by the interaction of multiple traits such as pods per plant (PN), pod weight (PW), pod length (PL), seeds per pod (SN/P), and 100-seed weight (100-SW) [34]. Variation in the number of pods was observed between the control and mutant lines during the 2023/24 season. The 2023/24 season was characterized by maximum temperatures ranging from 32.82 °C to 35.08 °C, while minimum temperatures varied from 16.10 °C to 19.37 °C. Additionally, the season experienced significant rainfall, with December receiving 73.5 mm. This was followed by lower amounts of 39 mm in January and 28 mm in February (Figure A1 and Figure A2). The combination of low rainfall and temperatures above 30 °C, along with recurring heatwaves during the flowering and podding stages, may have resulted in the decreased PW observed throughout the season [35]. This may have contributed to the increased flower abortion observed during the trial due to the limited soil moisture needed to sustain the plants through these critical stages [36]. Flower and pod abortion significantly limits the potential for achieving maximum grain yield [37,38]. Heat stress can lead to abnormal floral growth, altered flower structures, smaller flower sizes, and even completely sterile flowers [35]. These changes to flowers have been reported to lead to poorer pollination and fertilization, ultimately resulting in decreased fruit and seed production [39]. Despite a persistent heatwave during the study period, the Tswana-500Gy-53 mutant line produced more pods than the other mutant lines. This increase in pod count is likely due to enhanced floral development, as indicated by the higher number of flowers per node as well as the resilience exhibited by this mutant. The increase could also be attributed to physiological and biochemical changes induced by gamma irradiation. Cultivars developed through mutation may show elevated activity of antioxidant enzymes, which is essential for scavenging reactive oxygen species (ROS) generated under heat stress. This process helps protect the plant’s biological components from oxidative damage [40]. This is inconsistent with Bard et al. [41] and Khan et al. [42], who reported a low pod number in gamma-irradiated cowpea and pea, respectively. It was generally observed that the increased number of pods in mutant lines during the 2023/24 season was consistent with previous studies on peas, where the pod number increased with greater doses of gamma radiation [43,44]. Another study also reported a greater number of pods in cowpea mutant populations [12,31].

In addition to PN, SN/P is a critical determinant of grain yield. It has been reported that a variety with a higher SN/P can potentially yield more seeds per pod compared to one with fewer seeds per pod [45]. The observed high SN/P in some mutant lines aligns with the findings reported in mutant cowpea [46]. In contrast, a lower SNP was observed in the mutagenized groundnut, pea, and barley populations [47,48]. In this regard, variations in SN/P observed among the mutant lines across the two seasons could be attributed to induced changes in gene expression that regulate seed formation within the pod [49]. PW is another important factor in cowpea grain yield as it affects total seed weight. In contrast, 100-SW is strongly influenced by seed size and contributes significantly to the overall grain yield. Improvements in PW and 100-SW were noted in the 2023/24 season compared to the 2022/23 season, which led to an increase in GY for the 2023/24 season. Some mutant lines exhibited increased pod weight, seed weight, and grain yield, more likely due to induced genetic alterations affecting nutrient partitioning [50]. Mutations may have increased photosynthesis efficiency, subsequently redirecting plant resources toward pod and seed development, as well as enhancing nutrient uptake and transfer efficiency [51]. Furthermore, probable changes in hormone regulation, particularly auxins and gibberellins, may have contributed to increased seed size and higher yields [52]. Similarly, the high 100-SW in some mutants corroborates the findings observed in soybean and groundnut mutants [53]. Supanjani et al. [54] reported that low gamma doses less than 500 Gy contributed to a high 100-SW in green bean, which corroborates the current results from the 2023/24 season. Moreover, a lower 100-SW was reported in cowpea and lentil mutants, respectively [12,55]. The use of various crop species, with diverse developmental traits and responses to mutagenesis, as well as environmental variations between study sites, may all contribute to the reported inconsistency in results.

### 3.3. Heritability Estimates and Genetic Variability for the Agro-Morphological Traits

While yield parameters provide a measurable picture of crop performance, understanding the genetic parameters that drive these traits is critical for long-term crop improvement [56]. Selection based on GV and PV traits may provide benefits, and their phenotypic expression can accurately predict genotypic potential with minimal environmental influence [57]. Therefore, optimizing environmental conditions may be crucial in maximizing the expression of important traits for achieving improved yields [58]. Generally, this study revealed higher genetic variance (GV) and phenotypic variance (PV) for plant weight (PW) and grain yield (GY) in the 2023/24 season compared to the 2022/23 season. Specifically, the GV and PV for PW were 151.038 and 163.450, respectively, while for GY, they were 60.178 and 141.986. In contrast, during the 2022/23 season, the GV and PV for PW were only 3.556 and 7.483, respectively, and for GY, they were 5.201 and 61.888. This suggests a strong environmental influence on the expression of these traits across the two seasons [58]. This could be attributed to environmental factors, including recurring heat stress in 2023/24 and shorter day lengths, as well as decreasing temperatures in the 2022/23 season. The 2022/23 season experienced maximum temperatures ranging from 30 °C to 25 °C and minimum temperatures from 16.49 °C to 0.3 °C (Figure A1). As temperatures decreased, plants had to flower earlier, resulting in a smaller number of days to maturity. This, in turn, limited pod development and seed filling, which are critical factors in cowpea production.

Generally, a high coefficient of variation indicates greater potential for selecting desirable traits. In contrast, a low coefficient of variation highlights the need to generate variability and select for a specific environment. A slight variation was observed between the genetic coefficient of variation (GCV) and the phenotypic coefficient of variation (PCV) for most specific traits, except for GH (18.122; 38.553), PN (6.688; 28.277), and GY (5.676; 19.580). The minor variation noted may indicate a limited environmental influence on these traits, whereas the substantial difference between the GVC and PVC observed for GH, PN, and GY suggests a significant environmental impact on their expressed traits [59]. The slight variation in the observed GVC and PVC was consistent with previous studies. As such, selection based on these characteristics may be beneficial, and their phenotypic expression can accurately predict genotypic potential with minimal environmental impact [57]. Therefore, optimizing environmental conditions may be crucial in maximizing the expression of these traits and achieving improved yields [58]. In general, a high coefficient of variation suggests a greater scope for selection in favor of desirable traits. In contrast, a low coefficient of variation indicates the necessity for the generation of variability and selection [60]. Similar results with slight variations were reported in chickpeas [58,61], as well as in wheat and rice [60,62,63]. In contrast, a high GCV and PCV were observed for all the key agronomic traits in cowpea studies [64,65]. On the other hand, Goyal et al. [66] reported a decreasing trend in genetic variability parameters for mutant lines as gamma irradiation increases beyond 400 Gy. This is because gamma irradiation beyond 500 Gy has detrimental effects on plants, as it inhibits desirable characteristics compared to lower doses, which have stimulating effects [67].

The heritability of the traits also determines the efficiency of selection. Variation in broad-sense heritability alongside the GA percentage across seasons suggests inconsistency in the genetic and environmental influences on expressed traits [68,69]. This may indicate that the variation is due to the mutation of more than one locus [70]. The high H^2^ coupled with the high GA% recorded for PW in the 2023/24 season compared to the 2022/23 season indicates a strong genetic influence in determining phenotypic variation. This could imply that trait-based selection in 2022/23 would be less efficient in achieving genetic improvement than in the 2023/24 season, where genetics had a significant impact on the expression of the traits [71]. Most of the traits exhibited low-to-moderate heritability and genetic advance. GY had an H^2^ = 8.4% in 2022/23 and 42.4% in 2023/24. This inconsistency suggests challenges in reliably selecting traits and indicates that environmental factors significantly affected their expression. Environmental variability has a significant impact on heritability estimates and the efficacy of plant selection. It determines how quickly a trait responds to selection [72]. Environmental variations can have varying effects on gene expression, making it challenging to identify the genetic variations that are beneficial for adaptation. Uncertain environments can reduce the rate at which populations can adapt by decreasing genetic variation or by favoring different traits in different environments, leading to a less predictable selection trajectory [73,74]. This unpredictability can lead to changes in how traits are expressed across generations. For example, when there is high environmental variability, heritability tends to be low. However, this does not mean that genetics does not play a role in the trait. Rather, it may indicate that the significant environmental variability makes it more difficult to identify the genetic influences on the trait [65]. In general, heritability reflects the interaction between the genotype and environment, highlighting genetic potential and its expression across various environmental conditions [75]. As previously reported in cowpea and chickpea, this emphasizes the need to consider environmental factors and perform multi-season assessments when developing breeding methods. Similar findings have been reported in cowpea and chickpea [4,76,77].

### 3.4. Pearson Correlation Among the Agro-Morphological Traits

Beyond understanding heritability and genetic advance to drive breeding techniques, it is critical to determine which traits are most efficiently targeted for improvement. In this context, correlation analysis plays a vital role in determining the link between distinct yield attributes and grain yield [78]. The timing of emergence, as well as flowering time, can influence the allocation of plant resources and impact the plant’s overall growth pattern; hence, a positive and significant relationship was observed between DE and GH (r = 0.90 **). While not acting as a direct cause, the number of days to flowering can influence the overall length of the growth period, potentially affecting pod weight, which is reflected in the positive and significant relationship between DF and PW (r = 0.93 **). This could indicate that early flowering can lead to more pods compared to later flowering, as the early pods have more time to develop before resource allocation shifts to later flowers and pods [79]. Despite the insignificant positive relationship between GY and some of the traits (DE, GH, PN, PL, SN/P, and 100-SW), improving these traits would result in enhanced grain yield [32], as the selection of these positively correlated traits could be critical in improving cowpea production. However, the apparent differences in correlation coefficients in the assessed traits may be attributed to the pleiotropic effects of mutated genes [80]. Previous research contradicts our findings, indicating a strong association between 100-SW and GY in soybean and cowpea studies [31,81]. It can be inferred from the results that direct selection for grain yield could be carried out for traits that are positively correlated with yield. In addition to identifying trait correlations, analyzing genetic diversity within the population is critical for long-term success.

### 3.5. Cluster and Principal Component Analysis for Agro-Morphological Traits

Cluster analysis is useful for assessing genetic diversity as it displays genetic relatedness based on the various characteristics of the genotypes [82]. Despite the non-significant differences between the control and some lines, the control deviated from the others, confirming the alteration in the genetic makeup of the lines during the induced mutation [83]. This also could be because cluster analysis assesses the overall pattern of similarities and differences across several traits, where it can detect changes and multivariate differences that might not be apparent when analyzing individual traits separately [84]. Therefore, even slight, constant variations across some traits can result in distinct clustering. The results revealed the presence of genetic diversity among the mutants and the control. Diouf et al. [4] reported three cluster groups generated from a mutant population based on agro-morphological characteristics, suggesting the effectiveness of mutagenesis in inducing heterogeneous populations. This could result in significant divergence, which in turn could aid a breeding program [83]. Three clusters were also generated by Dabiré et al. [85] from a cowpea mutant population, indicating the presence of genetic diversity in the studied mutant population. Thus, the clustering of genotypes represents the extent of genetic variability, indicating better possibilities for genetic improvement in the crop through selection [86]. The PCA findings showed that the first three PCs accounted for 89.46% of the total variance. The distribution of the PCs indicated a significant level of quantitative trait variation [87]. The PCA demonstrated that Tswana-500Gy-31 was associated with improved GY, suggesting that this mutant line could be utilized in the breeding program as a parental line for the development of potential high-yield cultivars. Furthermore, the Tswana-300Gy-214, Tswana-400Gy-49, and Tswana-400Gy-85 lines are associated with a high SN/P, PW, and 100 SW, implying that selection for these traits will promote high GY [88]. Based on the distribution of mutagenized populations in the biplot, studies on cowpea mutants further revealed that mutagen doses, as well as the interaction of genotypes and their surrounding environment, generated significant variations in the expressed quantitative traits [89]. Research on PCA has reported that the mutants exhibit high-yield-related traits compared to the parents [90]. Overall, the development and comprehensive phenotyping of mutagenized lines offer opportunities to explore the genetic and genomic basis of these important agronomic traits. They also provide the unique ability to explore the genomic basis of traits in comparatively isogenic backgrounds, an ability which has recently been exploited in other legume crops [91]. The mutagenized lines characterized here could serve as the basis for future work to determine the precise genomic factors responsible for these important cowpea traits.

## 4. Materials and Methods

### 4.1. Plant Materials

Seven genotypes, which included a locally released cowpea variety ‘Tswana-Control’ and six M5-Tswana cowpea mutant lines obtained from a National International Atomic Energy Agency (IAEA) Regional Project, were used in this study. The mutant lines were generated using gamma radiation, induced at the following doses: 300 Gy, 400 Gy, and 500 Gy. The M5 lines were generated through consecutive single-seed descent (SSD) based on agronomic characteristics such as the number of days to emergence and flowering, 100-seed weight, pod weight, and number of seeds per pod (Table 5).

### 4.2. Experimental Site and Design

The experiment was conducted at NARDI Sebele fields (24°35′30.8″ S, 25°56′44.7″ E) during the 2022/23 and 2023/24 cropping seasons. The experiment was conducted using a randomized complete block design (RCBD) with three replications. Plots were leveled to a fine tilth before planting. There were seven plots per replication, with each plot consisting of three rows measuring 3 m in length, with an inter-row spacing of 0.75 m and an intra-row spacing of 0.50 m, resulting in a plot size of 3 m × 2.25 m. A border row was planted in the outer plots to minimize the border effect on the data. The space between the replicates was 1 m to allow for easy movement during record-taking and other management operations. Two seeds were planted per hill at a depth of 2–3 cm. Supplementary irrigation was carried out to field capacity during drought periods to support the plants’ growth and development. Thinning was carried out two weeks after emergence, leaving only one plant per hill to reduce competition and allow the plant to reach its full potential. Weeding was carried out manually with a hand hoe as needed throughout the growing season.

### 4.3. Data Collection

#### 4.3.1. Agronomic Traits and Grain Yield

Data was collected from all three rows in a plot. Agronomic traits associated with grain yield were recorded as described below throughout the experimental period. At physiological maturity, cowpea pods were harvested from healthy, disease-free plants. The pods were picked when they were dry but not shattered. Visual inspection was used to determine the dryness of the pods, and shattering was checked daily, primarily in the afternoon until late afternoon. The dry pods were picked from each plot from all the plants except for those along the boundary.


**
*Days to 50% emergence (DE):*
**


The number of days from the date of planting to when 50% of the seedlings had emerged above the soil surface.


**
*Days to 50% flowering (DF):*
**


The number of days from the date of planting to when 50% of the plants in the plot had their first flower initiation (anthesis).


**
*Growth habit (GH):*
**


The growth habit was scored as follows: 1 for indeterminate, 2 for semi-determinate, and 3 for determinate growth. The growth habit was based on the description of the International Board for Plant Genetic Resources Cowpea Descriptor [93].

***Number of pods per plant (PN***):

The number of pods per plant was determined by counting the total number of pods each plant had at harvest.


**
*Pod length (PL):*
**


The pod length was measured at physiological maturity from all the pods, and the average was calculated per plot/variety.


**
*Pod weight [PW (g)]:*
**


The weight of the harvested pods was determined using an electronic balance.


**
*Number of seeds per pod (SN/P):*
**


The number of seeds per pod from the weighed pods per plant was counted; hence, the average number of seeds per pod was determined.


**
*100 seed weight [100-SW (g)]:*
**


The seeds from the harvested plants per replication were bulked for each variety, and 100 seeds were counted and weighed for each.


**
*Grain yield (GY) [kg/h]:*
**


The seeds from the harvested plants were bulked per plot, weighed, and then used for the determination of grain yield for each variety, expressed as kg ha^−1^.

#### 4.3.2. Heritability Estimates for Induced Genetic Variability

The level of stability and expressivity of yield and yield-related traits for the mutant lines and their control was evaluated using the genotypic and phenotypic variances, genotypic coefficient of variation (GCV), phenotypic coefficient of variation (PCV), heritability (H^2^), and genetic advance (GA).


**
*Phenotypic (^δ^^2^ p) and genotypic (^δ^^2^ g) variances*
**


Phenotypic (^δ^^2^ p) and genotypic (^δ^^2^ g) variances were calculated according to [19] as follows:
(1)gδ2 = MSp − MSe/r
(2)pδ2=MSg/r and eδ2 = MSe/r
where

MSp, MSg, and MSe are the mean squares of phenotypes, of genotypes, and of error, respectively;

r = the number of replications.


**
*Phenotypic coefficient of variation (PCV) and genotypic coefficient of variation (GCV)*
**


The mean values were used for genetic analyses to determine the phenotypic coefficient of variation (PCV) and genotypic coefficient of variation (GCV), according to [94]:
(3)$$GCV\;(\%)\;=\;(\surd \sigma^{2}g/ẍ)\;*100$$
(4)$$PCV\%\;=\;(\surd \sigma^{2}g/ẍ)\;*100$$
where

σ^2^g (an estimate of genotypic variance) = (MSg − MSe)/r;

σ^2^p = an estimate of phenotypic variance;

r = the sample mean.


**
*Heritability estimate*
**


It was computed as per [95]:
(5)$$h^{2}\;(\%)\;=\;(\sigma^{2}g/\sigma^{2}p)\;*100$$
where

σ^2^g is the genotypic component of variance;

σ^2^p is the phenotypic component of variance.


**
*The expected genetic advance percentage*
**


The mean percentage was determined based on [96]:
(6)$$GA\;=\;k\;p\sigma .h^{2}$$
where

H^2^ = broad-sense heritability;

pσ = phenotypic standard deviation of the mean performance of the treated population;

K = 2.06, constant for 1% selection intensity.

### 4.4. Data Analysis

The analysis of variance and mean separation was statistically conducted using SAS 9.4 software (SAS Institute Inc. 2023. SAS/STAT^®^ 15.3 User’s Guide. Cary, NC, USA: SAS Institute Inc.), where significant means were compared using the Least Significant Difference (LSD) at a risk level of 5% (*p* < 0.05). Genetic variability, including the estimation of genetic parameters (genotypic and phenotypic variances, genotypic and phenotypic coefficients of variation, heritability, and genetic advance) was also analyzed using R-studio software version 4.4 embedded with the Variability Package (http://www.posit.co/, accessed on 17 October 2024). The relationship between agronomic characteristics was established using Pearson’s correlation coefficients. Cluster analysis using the Unweighted Pair Group Method with Arithmetic means (UPGMA) was performed on all measured agronomic traits to generate a dendrogram of the genotypes using SAS 9.4 software. The Euclidean method was used to calculate the similarity between data points. Principal component analysis (PCA) was performed using the mean value of each trait using R Studio software version 4.5.0 (URL http://www.posit.co/, accessed on 18 June 2025).

## 5. Conclusions

Crop science is working to develop novel crop mutant lines with improved growth, development, and high-yield potential. However, there are still gaps in achieving this goal due to limited and narrow variations within some crop species such as cowpeas [31]. This study investigated the impact of induced mutagenesis on Tswana cowpea lines, focusing on emergence, flowering, yield, yield-contributing traits, and genetic parameters and correlations. The results showed that some lines exhibited delayed emergence and flowering compared to the control, implying that the induced physiological and biochemical modifications impacted these key developmental processes. While the performance of mutants varied across various parameters, the Tswana-500Gy-31 mutant performed better than other mutant lines in terms of GY, potentially due to the activation of enzyme activities that promote seed production, size, and quality. The close similarity between phenotypic coefficients of variation and genotypic coefficients of variation across both seasons indicated minimal environmental influence on most observed traits. High (H^2^) coupled with a high GA% for PW in the 2023/2024 season, compared to the 2022/2023 season, mainly indicated additive genetic control of the expressed traits. Most of the traits had low H^2^ and GA% values, indicating restricted genetic diversity and/or high environmental effects. Despite this, a positive selection gain for GY, while also selecting for other desired qualities, remains a possibility. Moreover, cluster analysis based on agronomic traits demonstrated that mutagenesis successfully generated populations with diverse phenotypic differences. Tswana-300Gy-214, Tswana-400Gy-49, Tswana-400Gy-85, and Tswana-500Gy-53 demonstrated improved yield-related traits and grain yield, indicating that these genotypes can be further studied for adaptability and stability in various environments to be considered as potential parents for future breeding and crop improvement. These findings highlight the potential of induced mutagenesis to enhance genetic diversity in cowpeas for economically important traits, thereby benefiting future breeding and development initiatives. Furthermore, the cowpea mutants identified as superior could be exploited as potential parents to expand the current limited germplasm collection and develop cowpea genotypes with improved grain yield.

## Figures and Tables

**Figure 1 ijms-26-07543-f001:**
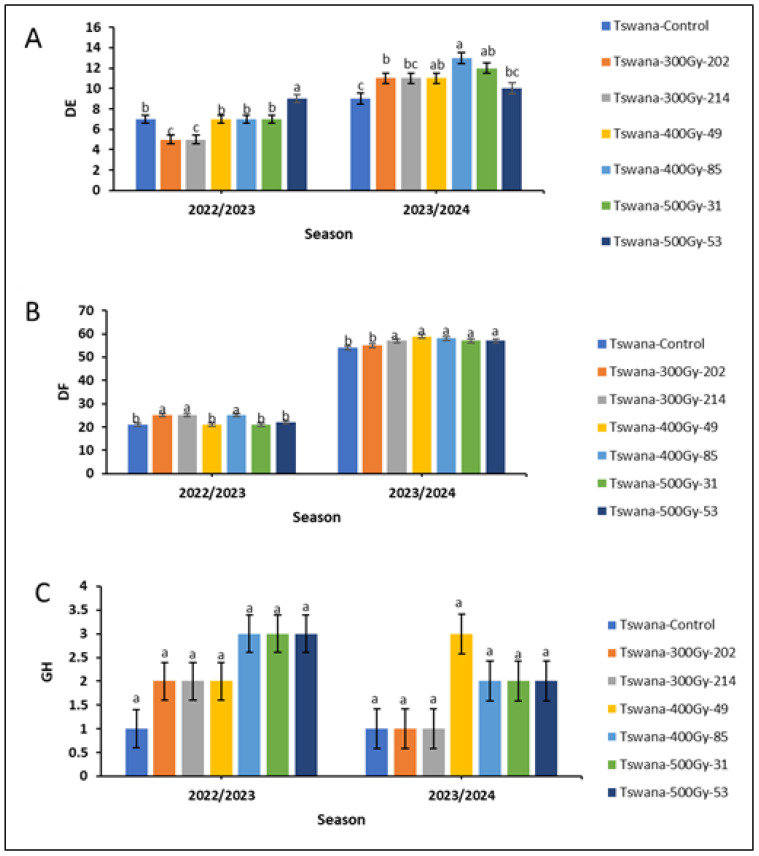
Number of days to 50% emergence (DE; (**A**)), number of days to 50% flowering (DF; (**B**)) and Growth habit (GH; (**C**)) for the M5 cowpea populations. Different letters within the season indicate significant differences at p ≤ 0.05. The error bars indicate the standard deviation of the mean.

**Figure 2 ijms-26-07543-f002:**
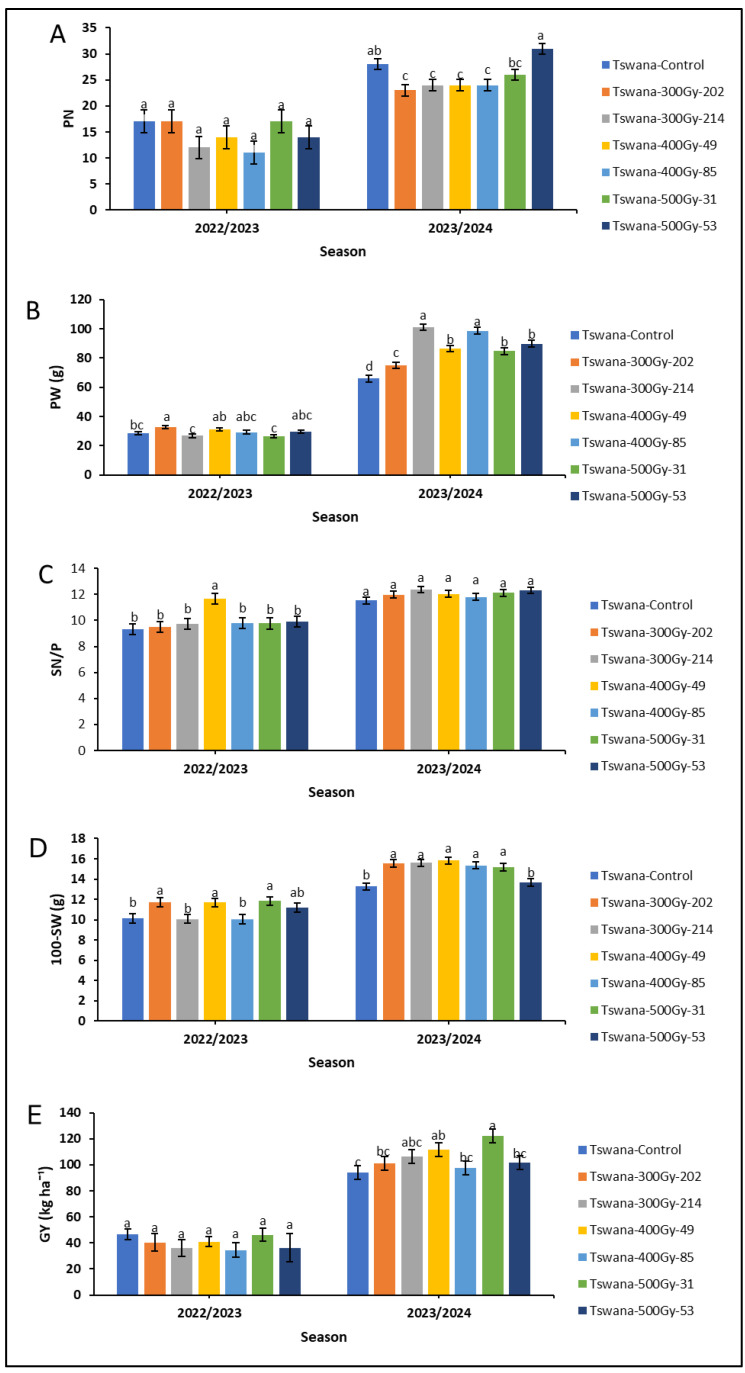
Yield and yield traits during the 2022/23 and 2023/24 seasons. Means with the same letters within the season indicate no significant difference at *p* < 0.05. The standard error indicates the standard error of the means. PN: pod number per pod (**A**); PW: pod weight (**B**); SN/P: number of seeds per pod (**C**); 100-SW; 100-seed weight (**D**); GY: grain yield (**E**).

**Figure 3 ijms-26-07543-f003:**
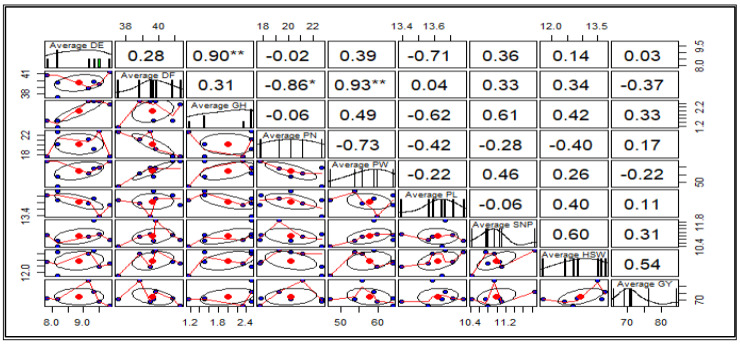
Pearson correlation coefficients for the M5 agro-morphological traits. Significance codes: ‘**’, 0.001; ‘*’, 0.05. Standard error of the mean (n = 3).

**Figure 4 ijms-26-07543-f004:**
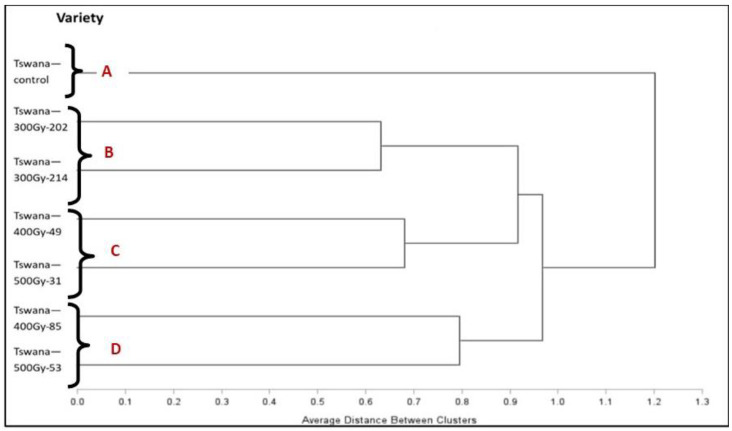
Dendrogram showing the similarity amongst the cowpea mutants evaluated based on their agro-morphological characteristics in the 2023/2024 season.

**Figure 5 ijms-26-07543-f005:**
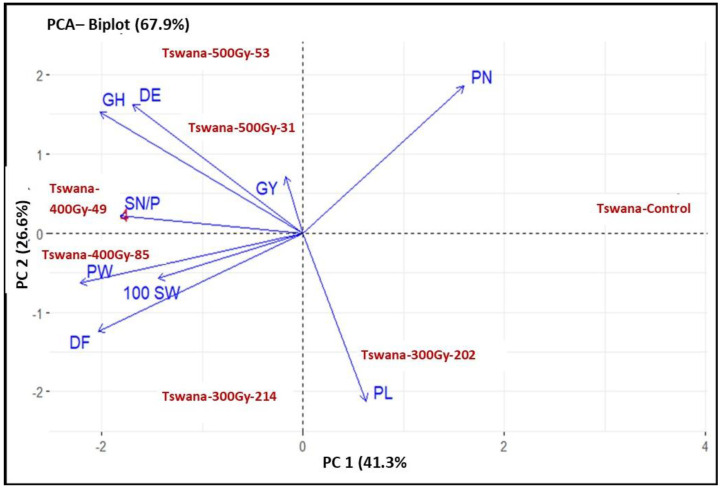
Principal component analysis (PCA) shows the associations among the genotypes and their traits. PC: principal component.

**Table 1 ijms-26-07543-t001:** Analysis of variance indicating mean square (MS) performance for agronomic traits of M5 cowpea population.

Source of Variation	DF	DE	DF	GH	PN	PW (g)	PL (cm)	SN/P	100-SW (g)	GY (kg ha^−1^)
Rep	4	0.857 ^NS^	2.643 ^NS^	0.286 ^NS^	4.091 ^NS^	13.757 ^NS^	0.073 ^NS^	0.621 ^NS^	0.404 ^NS^	48.736 ^NS^
Season	1	192.857 ***	12,036.214 ***	0.857 ^NS^	1326.298 ***	33,701.312 ***	0.087 ^NS^	44.620 ***	165.482 ***	44,141.451 ***
Var	6	3.690 ***	9.579 ***	2.056 ^NS^	26.809 *	212.427 ***	0.096 ^NS^	1.178 **	3.364 ***	210.144 *
Season*Var	6	4.246 ***	12.103 ***	0.135 ^NS^	17.175 ^NS^	267.676 ***	0.227 ^NS^	0.905 *	1.874 **	124.490 ^NS^
Error	24	0.635	1.698	0.536	9.558	8.175	0.236	0.301	0.474	69.248
CV %		8.996	3.287	36.596	15.289	4.964	3.566	4.994	5.321	11.463

Significance codes: ‘***’, 0.0001; ‘**’, 0.001; ‘*’, 0.05; NS, ‘not significant’ at *p* ≤ 0.05. Means followed by the same letter in the same column are not significantly different. Note: DF = degrees of freedom. VAR = variety. Standard error of the mean (n = 3).

**Table 2 ijms-26-07543-t002:** Overall genetic parameters of the M5 cowpea population during the 2022/23 cropping season.

Traits	GV	PV	GCV	PCV	H^2^	GA	GA %	MEAN ± SEM
DE	1.373	1.857	17.452	20.296	73.900	2.076	30.912	6.714 ± 0.402
DF	3.357	5.111	8.067	9.953	65.700	3.059	13.467	22.714 ± 0.765
GH	0.151	0.683	18.122	38.553	22.100	0.376	17.547	2.000 ± 0.421
PN	0.954	17.049	6.688	28.277	5.600	0.476	3.259	14.602 ± 2.316
PW	3.546	7.483	6.432	9.344	47.400	2.670	9.120	29.276 ± 1.146
SN/P	0.480	0.867	6.954	9.351	55.300	1.061	10.653	9.958 ± 0.359
100 SW	0.496	1.117	6.431	9.652	44.400	0.967	8.827	10.951 ± 0.455
Gy	5.201	61.888	5.676	19.580	8.400	1.362	3.389	40.178 ± 4.347

GV: genotypic variance; PV: phenotypic variance; EC: environmental variance; GCV: genotypic coefficient of variance; PCV: phenotypic coefficient of variance; H^2^: broad-sense heritability; GA: genetic advance; GA %: genetic advance as a percentage of the mean. Standard error of the means (n = 3).

**Table 3 ijms-26-07543-t003:** Overall variance parameters for the M5 cowpea population during the 2023/24 cropping season.

Traits	GV	PV	GCV	PCV	H^2^	GA	GA %	MEAN ± SEM
DE	0.849	1.635	8.378	11.624	51.900	1.368	12.437	11.000 ± 0.512
DF	2.738	4.382	2.925	3.700	62.500	2.695	4.764	56.571 ± 0.740
GH	0.222	0.762	25.382	47.001	29.200	0.524	28.237	2.000 ± 0.424
PN	7.336	10.357	10.481	12.454	70.800	4.696	18.172	25.841 ± 1.004
PW	151.038	163.450	14.302	14.878	92.400	24.337	28.322	85.929 ± 2.034
SN/P	0.014	0.229	0.988	3.981	6.200	0.061	0.505	12.020 ± 0.268
100 SW	0.934	1.261	6.476	7.524	74.100	1.713	11.483	14.921 ± 0.330
Gy	60.178	141.986	7.387	11.347	42.400	10.404	9.907	105.016 ± 5.222

GV: genotypic variance; PV: phenotypic variance: EC: environmental variance: GCV: genotypic coefficient of variance; PCV: phenotypic coefficient of variance; H^2^: broad-sense heritability; GA: genetic advance; GA %: genetic advance as a percentage of the mean; standard error of the means (n = 3).

**Table 4 ijms-26-07543-t004:** The eigenvalues and principal components of the 10 agronomic traits in cowpea genotypes.

Traits	PC1	PC2	PC3
**DE**	−0.342	0.410	−0.132
**DF**	−0.423	−0.312	−0.245
**GH**	−0.409	0.384	0.087
**PN**	0.323	0.468	0.066
**PW**	−0.449	−0.158	−0.240
**PL**	0.127	−0.536	0.333
**SN/P**	−0.369	0.056	0.294
**100 SW**	0.292	−0.144	0.518
**GY**	−0.036	0.179	0.623
**Eigenvalue**	3.17	2.39	1.94
**Variation%**	41.26	26.59	21.61
**Cumulative%**	41.26	67.85	89.46

**Table 5 ijms-26-07543-t005:** Agronomic characteristics of M5 cowpea mutant lines and control variety [92].

Cowpea Mutant Lines	Agronomic Characteristics
Tswana Control	Late maturing and low yielding
Tswana-300Gy-202	Early emergence and early flowering
Tswana-300Gy-214	Early emergence and early flowering
Tswana-400Gy-49	High seed weight
Tswana-400Gy-85	High pod and high seed weight
Tswana-500Gy-31	High seed weight, high yielding
Tswana-500Gy-53	High seed weight, high yielding

Gy: Gray, derived unit of ionizing irradiation dose; M5: mutant population; 202, 214, 49, 85, 31, and 53 represent codes for the mutant lines; 300 Gy, 400 Gy, and 500 Gy represent gamma irradiation doses.

## Data Availability

The data supporting this study are included within this article. Further inquiries concerning this article can be directed to the corresponding author.

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
