# Peer review of "Estimating Genetic Variability and Heritability of Morpho-Agronomic Traits of M5 Cowpea (Vigna unguiculata (L.) Walp) Mutant Lines"

_ijms, 2025, doi:10.3390/ijms26157543_

Round 1
Reviewer 1 Report
Comments and Suggestions for Authors
Dear Editor
Thank you for providing this opportunity to review the manuscript entitled “Estimating Genetic Variability and Heritability of Morpho-agronomic Traits of M5 Cowpea (Vigna unguiculata (L.) Walp) Mutant Lines”. After careful observations, I have following major comments for authors which need to be considered for this Journal.
Comments for authors
Abstract
Please provide the full forms of GH (Greenhouse) and PW (Panicle Weight) at their first mention.
Keywords
Add more specific and relevant keywords, such as Cowpea, mutation breeding, mutant lines
Change phenotypic and genotypic to “phenotypic variation” and “genetic variability”.
Introduction
The first sentence could be refined for clarity. The phrase “within a species or within population species” is redundant and slightly awkward. Consider revising to: “Genetic diversity refers to the extent of genetic variability within and among populations of a species.”
The transition from the general importance of genetic diversity to the specific focus on cowpea is well-intentioned but slightly abrupt. Adding a bridging sentence to smoothly guide the reader from the global context to the species-specific issue would strengthen the narrative.
The relevance of cowpea improvement to less-resourced communities is a strong justification for the study. However, the statement would benefit from supporting statistics or a reference that emphasizes cowpea's role in food security in these regions.
The introduction clearly identifies a problem; narrow genetic diversity in domesticated cowpea. This is a good foundation. However, the connection between this problem and the proposed solution (induced mutation) could be made more explicitly compelling.
The rationale for using induced mutation (gamma radiation) is well stated. You may consider briefly noting any advantages this method has over traditional breeding or modern molecular techniques to further justify its selection.
The sentence “...due to environmental influences and it is more useful...” should be split for clarity: “…due to environmental influences. It is more useful…”
Material & Methods
Section 4.1, the origin and mutagenesis process for the mutant lines are not detailed. Including information such as the type of mutagen used (e.g., gamma rays, EMS) and dose, if available, would improve scientific rigor and reproducibility.
There appears to be an inconsistency in table numbering: the text in section 4.1 refers to Table 1, but the corresponding table is labelled as Table 4. Please review and correct the table number.
Also, revise labelled Table 4 by reorganizing its layout into two distinct columns: one listing the mutant lines and the control variety, and the other detailing their corresponding agronomic characteristics. Change its caption to “Agronomic Characteristics of M5 Cowpea Mutant Lines and the Control Variety”.
"The subheadings '2.2. Experimental site' and '2.2.1. Field experiment' appear to be redundant, as they both refer to the same aspect of the study location and setup. I recommend consolidating these into a single heading, either retain 'Experimental site' and include all relevant details there, or rename it to 'Field experiment' if that better reflects the content."
Section 2.2, the phrase “Two seeds were planted per hill at a depth of 2-3 cm deep” is slightly redundant, consider rephrasing to “Two seeds were planted per hill at a depth of 2–3 cm.”
“Weeding was done manually using a hand hoe wherever it appeared”, consider rephrasing to “Weeding was performed manually using a hand hoe as needed throughout the growing season.”
The phrase “pods were harvested when they are dry but before they shatter” is important for reproducibility. Consider specifying how the dryness was assessed (visual inspection, moisture content) and how pod shattering was monitored or prevented.
For trait measurements, "please specify how many plants per plot were sampled and whether measurements were taken from all plants or a subset."
In statistical analysis, the use of UPGMA for cluster analysis is suitable. Consider specifying the distance measure or similarity index used (e.g., Euclidean, Manhattan) for clarity.
Provide reference for R-studio software version 4.4 and SAS 9.4 software.
Results
In table 1, give full form of “Var” in foot note.
In section 2.2, the comparison between seasons is well explained, but consider explicitly stating the statistical significance levels (e.g., “significantly fewer (P<0.05)”) when mentioning differences between seasons or varieties to reinforce the robustness of your findings.
The statement on growth habit (GH) could be emphasized more briefly since it shows no significant difference, e.g., “Growth habit (GH) showed no significant variation between seasons, indicating stability across environments.”
In figure 1, mention full form of GH, DF, DE in footnotes.
In section 2.3, A few sentences are lengthy and could be broken into shorter ones for easier reading. For example, “During the 2023/2024 season, the control variety had much greater PN (28 pods) than most of the varieties, with the exception of Tswana-500Gy-53 (31 pods), which had higher PN.” This could be simplified.
The section 2.3, presents valuable data but would benefit from clearer organization. Consider grouping results by trait (e.g., pods per plant, pod weight, seed number per pod, 100-seed weight, grain yield) and then discussing season-wise differences within each trait. This would enhance readability and flow.
In Table 2, there appear to be typographical or formatting errors in the data rows, with some numbers overlapping or misplaced (e.g., “2022/20 2023/20” instead of “2022/2023,” “62.500 3.059”). These should be carefully checked and corrected.
In table 2, The column headers should be clearly labelled, ideally with the trait names followed by each parameter for both seasons side-by-side (e.g., GV 2022/23, GV 2023/24, PV 2022/23, PV 2023/24, etc.). Consider using multi-level headers or merging cells for better structure, or clearly separate the data for the two cropping seasons (2022/2023 and 2023/2024) into distinct columns or sub-tables to improve clarity.
The mention of “Standard error of the mean (n=3)” is important but unclear in the context of the table 2. Please clarify which values have standard errors attached, and consider including the actual standard error values if relevant.
A few terms such as Genotypic Coefficient of Variation (GCV), Phenotypic Coefficient of Variation (PCV), heritability (H²), and genetic advance (GA) were introduced in the introduction; therefore, there is no need to provide their full descriptions again in the results, use only the abbreviations.
Discussion
When citing environmental influences, consider discussing soil moisture or nutrient status alongside heat stress, as these factors interactively affect growth and yield.
Clarify if broad-sense heritability estimates include genotype-by-environment interaction variance or if they are environment-specific.
Revisit the language in places to maintain clarity and avoid redundancy, especially in sections where multiple citations support similar findings.
Including specific numeric values or ranges of GCV, PCV, H², and GA% when discussing these parameters would improve precision.
This section is very dense and difficult to follow, reorganize this information under major subheadings.
Conclusion
The discussion on environmental influence vs. genetic control is insightful but slightly contradictory. The sentence mentioning high heritability coupled with high GA% “suggested the predominance of environmental impact” may need rephrasing, as high heritability usually indicates strong genetic influence. Clarify this point to avoid confusion.
The conclusion rightly emphasizes the potential of mutagenesis to increase genetic variability and improve important traits. It would strengthen the conclusion to explicitly propose future research directions.
References
Please follow journal guidelines and cross check all references with in-text citations.
Author Response
The attached document contains responses to all comments and recommendations. The English text was improved using Grammarly.

Reviewer 2 Report
Comments and Suggestions for Authors
I suggest the following changes:
Additional bibliography should be added in the introduction section, in order to attract the reader's interest in the topic.
Include more detailed environmental data (e.g., rainfall, temperature, soil characteristics) for the two seasons. This would help contextualize the observed seasonal effects on trait expression.
Despite claims of minimal environmental impact on most traits, key agronomic traits like grain yield (GY), pod weight (PW), and pod number (PN) showed significant variation across seasons. This seasonal variation complicates conclusions about genetic effects versus environmental factors.
Many traits exhibited low to moderate heritability and genetic advance (e.g., GY had H² = 8.4% in 2022/23 and 42.4% in 2023/24). This inconsistency suggests challenges in reliable trait selection and indicates that environmental factors significantly affected expression.
While the results and discussion sections imply limitations (e.g., environmental variability, low heritability for some traits), a dedicated paragraph acknowledging these directly would enhance scientific transparency.
The discussion could further connect findings to practical breeding programs—e.g., how specific mutant lines could serve as parents for hybrid development or in backcrossing schemes.
Improve data presentation with heatmaps or principal component analysis (PCA) to better visualize trait interrelationships and genotype performance trends.
The results are based on just two cropping seasons, with noticeable seasonal effects. This restricts the understanding of genotype × environment (G×E) interaction and long-term trait stability.
Data was collected from a single research station, meaning findings may not extrapolate to other agroecological zones due to environmental variability (e.g., soil type, microclimate).
Author Response
The attached document contains responses to all comments and recommendations. The English text has been improved using Grammarly.

Round 2
Reviewer 1 Report
Comments and Suggestions for Authors
The authors have adequately addressed all of my comments; therefore, I have no additional remarks.
Author Response
Dear Reviewer
Thank you for the insightful comments and recommendations which have improved our maunscript. Your guidance is much appreciated.

Reviewer 2 Report
Comments and Suggestions for Authors
I suggest the following changes:
Grain yield (GY) showed very low heritability (8.4%) in one season — the authors should discuss more thoroughly how environmental variation limited genetic selection potential.
Additional references to recent genomics-enabled mutation breeding in legumes would enhance the depth, especially in the discussion.
Comments on the Quality of English LanguageImprove language clarity and grammar across sections (especially Results and Discussion).
Author Response
Dear Reviewer
Your comments have been very helpful in improving the manuscript. Your comments and recommendations are much appreciated.
